# XRN2 suppresses aberrant entry of tRNA trailers into argonaute in humans and Arabidopsis

**Briana Wilson**[1], **Zhangli Su**[2], **Pankaj Kumar**[1], **Anindya Dutta**[1,2]*

**1** Department of Biochemistry and Molecular Genetics, University of Virginia School of Medicine, Charlottesville, Virginia, United States of America, **2** Department of Genetics, University of Alabama, Birmingham, Alabama, United States of America

* duttaa@uab.edu

**Data Availability Statement:** All small RNA sequencing data has been deposited in the Gene Expression Omnibus (GEO) database under accession code GSE184124 and can be accessed

## Abstract

MicroRNAs (miRNAs) are a well-characterized class of small RNAs (sRNAs) that regulate gene expression post-transcriptionally. miRNAs function within a complex milieu of other sRNAs of similar size and abundance, with the best characterized being tRNA fragments or tRFs. The mechanism by which the RNA-induced silencing complex (RISC) selects for specific sRNAs over others is not entirely understood in human cells. Several highly expressed tRNA trailers (tRF-1s) are strikingly similar to microRNAs in length but are generally excluded from the microRNA effector pathway. This exclusion provides a paradigm for identifying mechanisms of RISC selectivity. Here, we show that 5′ to 3′ exoribonuclease XRN2 contributes to human RISC selectivity. Although highly abundant, tRF-1s are highly unstable and degraded by XRN2 which blocks tRF-1 accumulation in RISC. We also find that XRN mediated degradation of tRF-1s and subsequent exclusion from RISC is conserved in plants. Our findings reveal a conserved mechanism that prevents aberrant entry of a class of highly produced sRNAs into Ago2.

## Author summary

Gene expression is a tightly regulated process. One way gene expression is controlled is through small RNA binding to mRNAs within the RNA induced silencing complex (RISC), which subsequently reduces protein production. Although this process is canonically thought to be mediated by a small RNA class known as microRNAs, it is now clear that the RISC-associated small RNAome is more complex. We have shown that some, but not all, small fragments of transfer RNAs (tRFs) are able to interact with RISC. The mechanism by which some small RNAs enter RISC and not others is unclear. Here we provide evidence that tRFs derived from tRNA trailers (tRF-1s) have very short half-lives mediated by XRN2 degradation, which prevents tRF-1 accumulation in RISC in both humans and plants. Our study highlights a conserved mechanism in which a class of abundant small RNAs are not permitted to enter RISC under basal conditions. These findings highlight

at https://www.ncbi.nlm.nih.gov/geo/query/acc.
cgi?&acc=GSE184124.

**Funding:** This research was supported by the NCI
Cancer Center Support Grant 5P30CA044579; NIH
grant R01GM146756 and R01CA060499 (to A.D.)
which paid the salary for AD, ZS, BW; NIH NCI F30
Grant 1F30CA254134 (to B.W.) which helped to
pay the salary for BW; NIH K99 Grant CA259526
(to Z.S.) which helped to pay the salary for ZS. The
funders had no role in study design, data collection
and analysis, decision to publish, or preparation of
the manuscript.

**Competing interests:** The authors have declared
that no competing interests exist.

the complexity and tight control of RISC and have implications for both tRF and micro-
RNA biology.

## Introduction

microRNAs (miRNA) are small noncoding RNAs around 22 nucleotides long that downregu-
late gene expression by base pairing to target RNAs within argonaute (Ago) proteins, mainly
Ago2 [1,2]. This functional complex is known as the RNA-induced silencing complex (RISC).
miRNA function is highly regulated, as miRNAs control many different homeostatic processes
[3]. An important level of regulation is at the point of Ago entry. One regulatory mechanism
for Ago entry that has been proposed is that Dicer-2, which is involved in the biogenesis of siR-
NAs from long double stranded precursors, has an integral role in loading siRNA into RISC in
flies [4]. However, the human genome encodes a single Dicer gene and it is dispensable for
RISC loading in mammalian cells [5]. Supporting the latter finding, we and others have identi-
fied DICER-independent transfer RNA fragments, or tRFs, associated with Ago proteins
[6–8].

tRFs have been implicated in a variety of cellular and disease processes, including transla-
tion inhibition, nascent RNA silencing, cancer pathogenesis and progression, and viral infec-
tion [7,9–18]. tRFs arise from tRNAs, which are essential components of the cellular
translation machinery. All tRNAs are transcribed by RNA polymerase III (Pol III) as a longer,
precursor tRNA containing a leader sequence, a trailer sequence, and for some tRNAs, introns.
The trailer sequence contains an important termination signal, whereby Pol III terminates fol-
lowing poly-uridine synthesis [19,20]. These precursor sequences must be removed in order to
produce a mature tRNA. In mammalian cells, this maturation involves endonucleolytic cleav-
age by RNaseP for removal of the leader sequence, RNaseZ for removal of the trailer sequence,
and the tRNA splicing endonuclease (TSEN) complex for the removal of introns [21–23]. Each
of these steps produces tRF-leaders, tRF-1s, and intron-tRFs, respectively, with tRF-1s being
the most characterized. tRF-1s are generally between 15 and 22 nucleotides long, with variable
3' end due to variable Pol III termination (Fig 1A). In addition, other tRF types arise from
mature tRNA sequences and are named based on the end of the mature tRNA from which it
arises, with tRF-5s originating from the 5' end and tRF-3s originating from the 3' end (Fig 1A).
Miscellaneous tRFs (misc-tRFs) can arise from internal sequences of tRNAs, and also include
intron-tRFs (Fig 1A). Due to similar size to miRNAs, many tRFs are ideal substrates for cap-
ture by classical small RNA (sRNA) sequencing library preparation approaches [24].

Although tRF-1s were one of the first tRFs characterized, few biological roles for tRF-1s
have been identified to date [15]. One specific tRF-1, tRF-1001, has been found to play an
important role in cellular proliferation [25]. It was noted that tRF-1001 was highly abundant,
but had no miRNA-like function. This finding was validated in a separate study that found
tRF-1s unable to enter Ago2 and repress gene expression [26]. This is in contrast to tRF-5s and
tRF-3s that associate with Ago proteins and repress gene expression in a variety of systems and
organisms [6,8,24,26,27], suggesting that tRFs are not generally excluded from entering the
miRNA effector pathway. Therefore, it is likely that specific cellular mechanisms prevent the
accumulation of tRF-1s into RISC.

Interestingly, addition of an RNA antisense to tRF-1s licenses tRF-1 entry into RISC and
imparts miRNA function [26]. Since addition of an antisense RNA may lead to increased sta-
bility, one possibility is that tRF-1s are absent from RISC due to instability. Here we find that
XRN2 degrades tRF-1s, rendering this class of small RNAs highly unstable relative to other

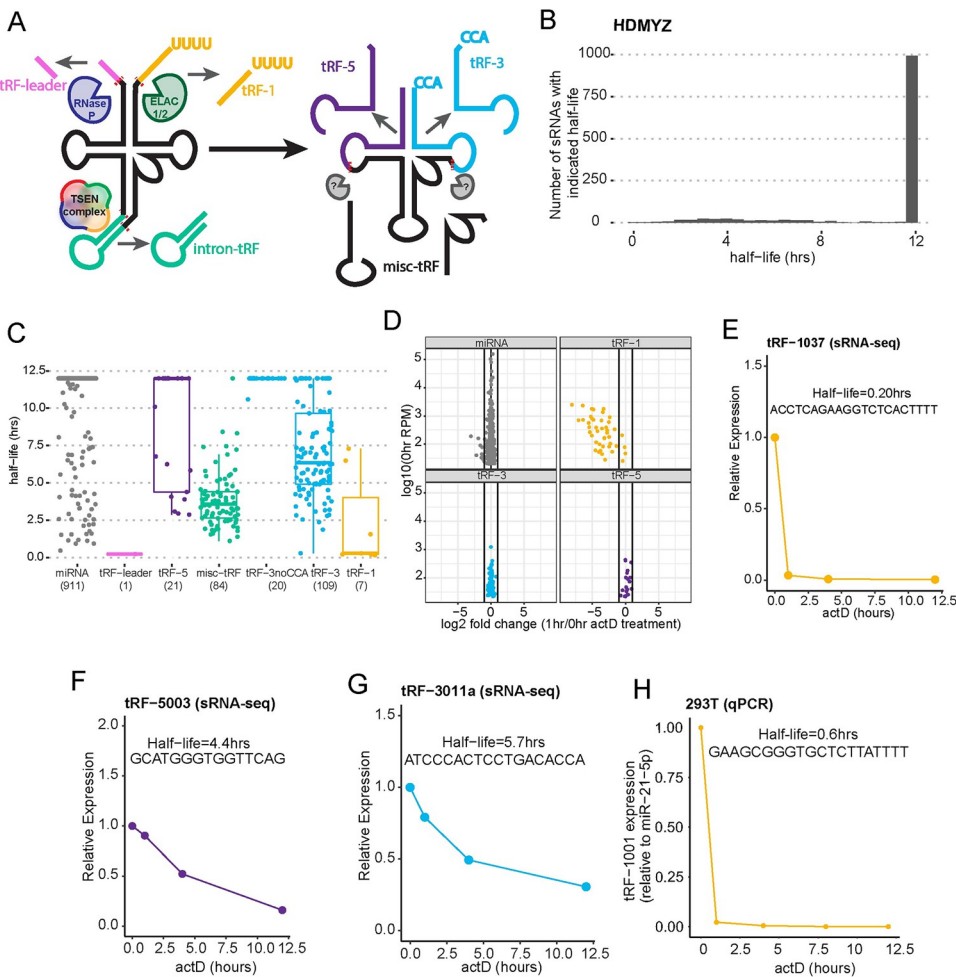

**Fig 1. tRF-1s are highly unstable.** A) Schematic of tRNA fragment (tRF) biogenesis. Precursor tRNAs are processed by RNase P, ELAC1/2, and the TSEN complex to give rise to tRF-leaders, tRF-1s, and intron-tRFs, respectively. It is unknown what enzymes produce the majority of tRF-5s and tRF-3s from mature tRNAs. B) Distribution of small RNA half-life in HDMYZ cells. Plotted are the number of short RNAs with indicated half-life. C) Distribution of small RNA half-lives by small RNA class. In parenthesis on the x-axis are the number of small RNAs plotted for each class. D) Levels of miRNAs and tRFs in HDMYZ cells at 0 and 1 hour actinomycin D treatment. E-G) Examples of degradation kinetics for the three major tRF classes in HDMYZ cells (sRNA-seq). H) Validation of tRF-1 instability in 293T cells by size selection and RT-qPCR. Cells were treated with actinomycin D for the indicated amount of time.

tRF classes. When XRN2 is depleted, tRF-1s are stabilized and associate with Ago2. We find that XRN-mediated degradation of tRF-1s and subsequent exclusion from RISC is conserved in plants, suggesting the evolutionary importance of this degradation pathway.

# Material and methods

## Cell culture, siRNA transfection, and actinomycin D treatment

293T cells, human embryonic kidney cells transformed by adenoviral and SV40 oncogenes, were maintained in DMEM with L-glutamine, 10% fetal bovine serum, and 1% penicillin/streptomycin. 293T was obtained from ATCC (ATCC #CRL-3216) and were grown in humidified incubators with 5% $CO_2$ at 37°C. For duplex siRNA transfections, cells were first reverse transfected, then forward transfected 24 hours later using RNAiMax following the RNAiMax

protocol. siCon sequence is 5′-CGUACGCGGAAUACUUCGA-3′/5′-UCGAA-GUAUUCCGCGUACG-3′, siXRN1 5′-AAUUAUUCCUCAAUGAUAGUG-3′/5′-CUAUC AUUGAGGAAUAAUUAC-3′, siXRN2 5′-UUUACAAAAGCUUAAUUCCAA-3′/5′-GGA AUUAAGCUUUUGUAAAGC-3′. Wildtype and catalytic dead XRN2 293 cells were a kind gift from the David Bentley laboratory [28].

## Northern blot

Total RNA was collected using Trizol following the manufacturer's protocol. 30ug RNA was run on a 15% TBE-Urea gel and stained with SYBR gold to determine equal loading. RNA was then transferred onto Amersham HyBond-N+ membrane and probed using ExpressHyb hybridization solution following manufacturer instructions. Specific probes for northern blot were: 5′-BIO-AAAATAAGAGCACCCGCTTC-3′ for precursor tRNA SerTGA and tRF-1001 detection, and 5′-BIO-AAACGAGGTAACTCCGGAGCACA-3′ for precursor tRNA CysGCA and tRF-1015 detection.

## RT-qPCR

For small RNA detection by RT-qPCR, total RNA was collected using Trizol following the manufacturer's protocol. 30ug RNA was run on a 10% TBE-Urea gel, stained with SYBR gold, and small RNAs between 15-50nt were excised from the gel. Small RNA was eluted overnight in elution buffer containing 1M Tris-HCl (pH 7.5), 2.5M sodium acetate, 0.5M EDTA, and 10% SDS. RNA was collected using phenol-chloroform extraction. RT-qPCR was performed using the miScript primer assay kit (Qiagen) following manufacturer instructions using tRF-1001 (5'-GAAGCGGGTGCTCTTATTTT-3') and miR-21-5p (5'-TAGCTTATCAGACTGA TGTTGA-3') probes for qPCR.

## Small RNA-seq library preparation and analysis

Small RNA-seq library preparation was similar to previously described [29] with addition of spike-in. Briefly, 1 ul (500 ul total volume) of Qiaseq miRNA library QC spike-in (Qiagen #331535) were mixed with 1 ug total RNAs before library preparation. The resulting libraries were size selected for 15–50 nt inserts and pooled for sequencing on Illumina NextSeq500 according to validated standard operating procedures established by the University of Virginia Genome And Technology Core, RRID: SCR_018883.

HDMYZ cells are an acute myeloid leukemia cell line. Small RNA sequencing data from HDMYZ cells was downloaded from GSE46968. There are three biological replicates for each actinomycin D time point. For actinomycin D small RNA sequencing, if the small RNA decreases to a size below the sequencing library cutoff, these small RNAs will not be cloned and counted. This does not affect the conclusions because the shortening is part of the degradation processes. Arabidopsis small RNA sequencing data was downloaded from GSE133461. There are two biological replicates for each total small RNA sequencing library and three biological replicates for each Ago RNA immunoprecipitation sequencing library. Small RNA sequencing data was first trimmed to remove adapters using cutadapt [30]. Only reads longer than 14 nucleotides were kept for mapping. Small RNAs were mapped using unitas 1.7.5 [31] allowing up to one mismatch. Isoforms of small RNAs were collapsed. Unnormalized mapped reads for miRNA and tRFs were imported into R and differential expression analysis was performed using DESeq2, with batch correction where necessary [32]. We required that at least 10 reads per small RNA across all samples to pass an expression threshold for differential expression analysis. Small RNAs were considered highly expressed if present at 50 normalized counts or higher.

ENCODE XRN2 eCLIP bigwig files (hg38) were downloaded from the ENCODE website (experiments: ENCSR657TZB, ENCSR655NZA, files: ENCFF323ABS, ENCFF305SDT, ENCFF782AAU, ENCFF723HML) [33,34]. Tracks were either screenshots from the UCSC genome browser, or deeptools was used to create metagene plots for tRNAs and miRNAs [35].

## Half-life analysis

To determine half-life in HDMYZ cells at a sequence level, we first performed decay profile normalization on normalized counts as described in Sorenson et al 2018 [36]. Briefly, all normalized counts were normalized to the zero time point (T0). Then, stable small RNAs were identified as those RNAs that had an apparent increase over time. This happens because total cellular RNA decreases with transcription inhibition over time making stable small RNAs appear to increase if normalizing by total RNA. A decay factor was calculated for each time point. The decay factor is the average T0 normalized count for each time point for the stable small RNAs. All small RNAs counts were then divided by their respective decay factor for each time point. Decay profile normalized counts per small RNA were fit using the following equation:

$$A(t) = A_0 e^{(-k^* t)}$$

Where A is abundance, $A_0$ is the initial normalized count at time zero (steady state abundance), t is time, and k is the decay rate. K was then used to find the half-life using the following equation:

$$half - life = ln\,(2)/k$$

For plotting purposes, small RNAs with half-lives predicted to be beyond 12 hours (the duration of the experiment) had their half-lives set to 12 hours. Phastcons scores were obtained using the bioconductor package phastCons100way.UCSC.hg19.

## Ago2 RNA-immunoprecipitation

Stable Flag-HA-Ago2 293T cells were generated by 5-day puromycin selection after transduction with lentivirus-containing supernatant. Lentivirus expression plasmids pLJM1-Flag-HA-Ago2-WT (Addgene #91978) was a gift from Joshua Mendell [37], pMD2.G (Addgene #12259) and psPAX2 (Addgene #12260) were gifts from Didier Trono. Flag-HA-Ago2 was immunoprecipitated using either Flag-M2 magnetic beads from Sigma or anti-MYC mouse antibody 9E10 as a negative control in RIP lysis buffer containing 50mM Tris pH 7.4, 150mM NaCl, 0.5% Triton X-100, and RNase and proteinase inhibitors. Complexes were immunoprecipitated overnight at 4 degrees Celsius. Complexes were washed three times with RIP lysis buffer and RNA eluted with Trizol. Small RNAs were either processed for small RNA sequencing or northern blotting. Northern blotting was performed as previously described [6].

## Statistical analysis

Statistical analyses are described in figure legends.

## Results

### tRF-1s are highly abundant and unstable by half-life analysis

To test whether endogenous tRF-1s are inherently unstable, we determined half-life of each small RNA sequence in HDMYZ cells following actinomycin D (actD) treatment [38]. The small RNA sequencing data captured 1260 unique sequences (S1 Table). Most small RNAs

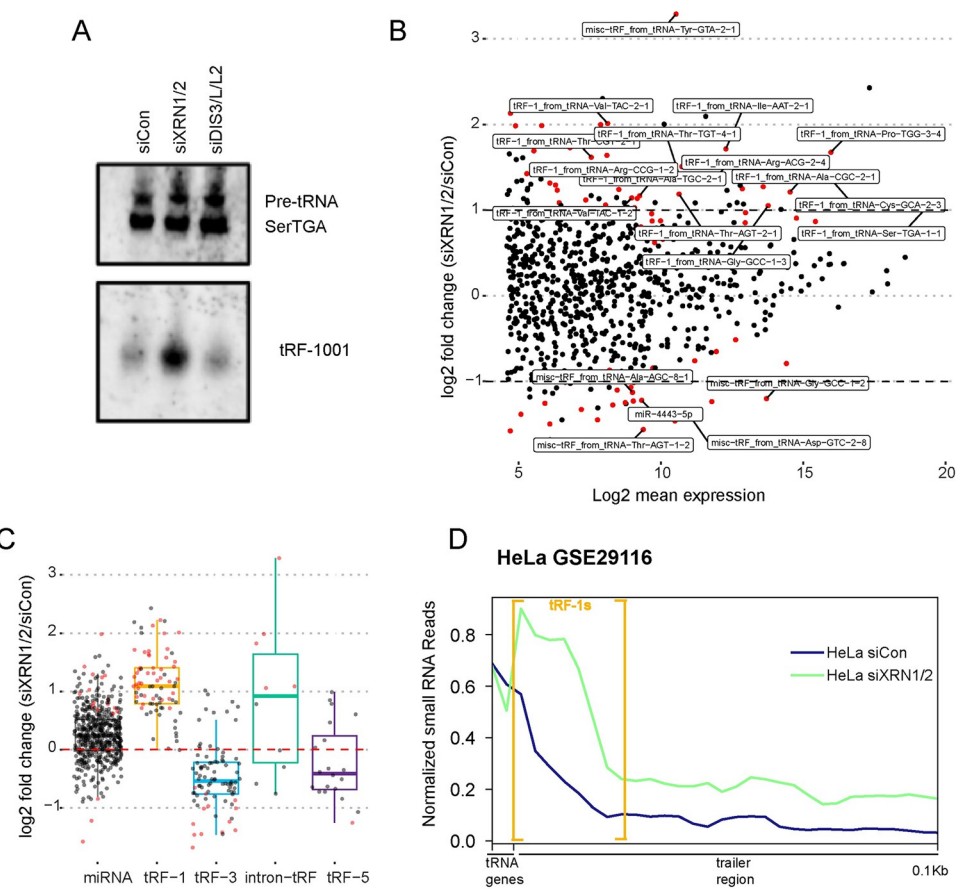

**Fig 2. XRN1/2 depletion increases tRF-1 expression.** A) tRF-1001 levels by Northern blot after co-knockdown of DIS3, DIS3L, DIS3L2, or co-knockdown of XRN1 and XRN2 in 293T. B) MA plot of small RNA alterations following co-knockdown of XRN1 and XRN2 (siXRN1/2). Red points are statistically significant. Differential expression determined by DESeq2. C) Boxplot of small RNA alterations following siXRN1/2 by class. Red points are statistically significant. D) Metagene plot of tRNA trailer region in HeLa cells after siXRN1/2 (data reanalyzed from GSE29116). Shown is the 3′ end of the tRNA genic region. The trailer region shown begins just downstream of the 3′ end of the tRNA gene and ends 100 bases downstream. tRF-1s are generally 15–22 bases long and are shown in yellow brackets.

were stable for the duration of the experiment (12 hours) (Fig 1B and S1 Table). Overall length distribution of small RNAs was unchanged by actD treatment (S1A Fig). The large number of small RNAs with half-lives of at least 12 hours is primarily driven by miRNAs (Fig 1C), while there are significantly fewer small RNAs that have half-lives of less than 12 hours. Surprisingly, there were very few tRF-1s detected after 12 hours of actD, and the seven tRF-1s that were detected have very short half-lives (median half-life of 0.28 hours) compared to stable miRNAs (median half-life greater than 12 hours), tRF-3s, and tRF-5s (Fig 1C). There was no clear pattern for tRF-1s that remained at 12 hours. The tRF-1s ranged from 14 to 25 nucleotides in length, with an average length of 21 nucleotides. The tRF-1s arose from various parental precursor tRNAs: tRNA-Leu-TAG-3-1, tRNA-His-GTG-1-7, two 3′-poly-U tail isoforms from tRNA-Val-TAC-1-2, tRNA-Thr-AGT-1-2, tRNA-Asn-GTT-2-2, and tRNA-Ser-GCT-4-1. Therefore, it is currently unclear what feature contributes to these tRF-1s relative increased stability in this cell line. The miRNA half-life is consistent with what is in the literature, which is more than 24 hours for the guide strand [39]. There was only one tRF-leader that was captured in this small RNA sequencing library. The size ranges of tRF-leaders are less than standard

small RNA sequencing size selection cutoffs (~15nts) and so are not accurately captured by sRNA seq. Therefore, in this paper we have mainly focused on other highly abundant tRFs.

Recognizing that there are far more tRF-1s at steady state than are present at 12 hours of actD, we next compared the one-hour and zero-hour time points in HDMYZ cells treated with actD to get a more global view of the turnover kinetics of tRF-1s. tRF-1s were selectively depleted by one hour with a median log2 fold change of -3.425 (Fig 1D). miRNAs, tRF-5s, and tRF-3s changed very little by one-hour, with a median log2 fold change of 0.0409, 0.273, and 0.0573, respectively. Note that tRF-1s have the highest median expression among tRFs and miRNAs at steady state or zero-hour actD treatment (median expression of 388.61 normalized counts), and is depleted the most by 12 hr. (S1B Fig). We selected three annotated tRFs [40] in each known tRF class to analyze turnover kinetics for the duration of the actD treatment. The turnover kinetics of tRF-1037 (Fig 1E) has rapid, single step turnover kinetics compared to tRF-5003 and tRF-3011a (Fig 1F and 1G). To our knowledge, this is one of the shortest half-lives for an abundant sRNA detected in human cells.

To determine if tRF-1s are unstable across different cell types, we treated 293T cells with actD for up to 12 hours and measured tRF-1 and miRNA levels by RT-qPCR. We analyzed the turnover kinetics of tRF-1001 because it is the most abundant tRF-1 in 293T and has been previously shown to have biological function [25]. Indeed, tRF-1001 is highly unstable in 293T as well (Fig 1H). These data suggest that tRF stability is differentially regulated based on tRF class and that instability of tRF-1 is not cell-type specific.

## Co-depletion of XRN1 and XRN2 increases tRF-1 levels globally

tRF-1 biogenesis is well understood [21], however, tRF-1 decay is uncharacterized. In order to identify the ribonuclease(s) (RNases) responsible for the very rapid tRF-1 turnover, we examined RNases in the XRN and DIS3 families, because they have been implicated in turnover of some sRNAs [41–43]. Knockdown of the XRN but not DIS3 families showed the tRF-1001 level was increased by Northern blot (Fig 2A). These data pointed towards the 5′-3′ exoribonuclease 1 and 2 (XRN1 and XRN2) as being putative tRF-1 regulators. To determine if the effect of XRN1 and XRN2 on tRF-1s was global, we performed small RNA sequencing after co-transfection of siXRN1 and siXRN2 (siXRN1/2) in 293T cells. RNA was size selected to capture small RNAs shorter than 50 nucleotides, which we expect to cover the majority of annotated tRF-1s. Consistent with the hypothesis that XRN1/2 degrades tRF-1s, several tRF-1s were significantly increased following siXRN1/2 (Fig 2B and S2 Table). Analysis of small RNA changes by subclass revealed specific increases in intron-tRF and tRF-1 classes at a global level following XRN1/2 depletion (Fig 2C). Consistent with XRN1/2 regulating tRF-1s in several cell types (and species, as discussed later), reanalysis of small RNA sequencing after knockdown of XRN1/2 in HeLa cells also shows an increase in reads from the trailer region, consistent with an increase in tRF-1 levels (Fig 2D) [44].

## XRN2 degrades tRF-1s

Since XRN1 and XRN2 may regulate tRF-1s at the level of production or degradation, we directly measured pre-tRNA and tRF levels by Northern blot in XRN1/2 depleted conditions before and after treating cells with actinomycin D (actD) for one hour. tRF-1s were clearly increased after co-depletion of XRN1 and XRN2, without an increase of its precursor, pre-tRNA SerTGA, indicating that XRN1 and/or XRN2 are involved in the degradation of tRF-1s (Fig 3A).

We initially knocked down XRN1 and XRN2 together due to concerns that one enzyme could compensate for the other [45], but proceeded to knock down each exoribonuclease

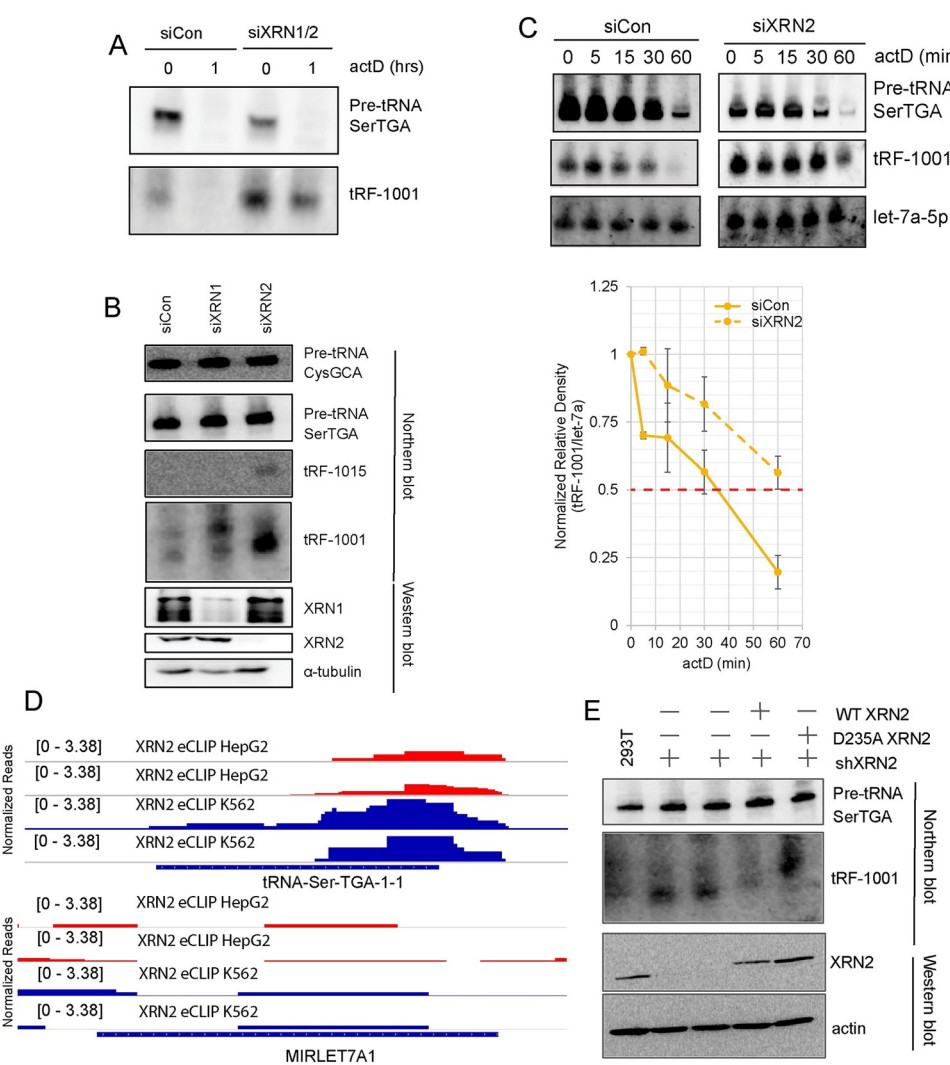

**Fig 3. XRN2 degrades tRF-1s.** A) Northern blot analysis of tRF-1001 and parental pre-tRNA SerTGA. Cells were co-depleted of XRN1/2, then treated with actinomycin D for 1 hour. B) Northern blot analysis of tRF-1001, tRF-1015, and parental pre-tRNAs following depletion of XRN1 and XRN2 separately. Western blot analysis showing knockdown efficiency below. C) Northern blot analysis of degradation kinetics for tRF-1001, parental pre-tRNA, and let-7a-5p after XRN2 knockdown. Quantitation of northern blot below (n = 2). Error bars represent standard error. D) XRN2 eCLIP genome browser tracks for tRNA-Ser-TGA-1-1 (tRF-1001 parental pre-tRNA) and let-7a1 in HepG2 and K562. The tRNA trailer begins just downstream of the 3′ end of the tRNA-Ser-TGA-1-1 gene. E) Northern blot of tRF-1001 and parental pre-tRNA following doxycycline induced expression of wildtype or catalytic dead (D235A) XRN2 in 293 cells.

separately to determine which exoribonuclease is responsible for tRF-1 degradation. Depletion of XRN2 alone was sufficient to increase tRF-1001 and tRF-1015 expression (Fig 3B). Furthermore, depletion of XRN2 stabilizes tRF-1001, tRF-1007, and tRF-1015 also on Northern blots (S3A Fig). We conducted a more refined actD time course to assess alterations in tRF-1 half-life when XRN2 is depleted. Depletion of XRN2 resulted in a half-life about twice as long as control knockdown (Fig 3C, including the quantitation from two separate experiments).

To determine if XRN2 can directly bind tRF-1s, we analyzed enhanced crosslinking and immunoprecipitation (eCLIP) from ENCODE [46]. Since XRN2 is a 5′-3′ exoribonuclease, we expect to find eCLIP reads near the 5′ end of tRNA trailer sequences and/or 3′ end of tRNAs.

Indeed, we found XRN2 eCLIP reads at the tRNA SerTGA-1-1 trailer sequence, the sequence for tRF-1001, in both K562 and HepG2 cells (Fig 3D). The number of XRN2 eCLIP reads was higher for tRF-1001 compared to let-7a-1, the latter showing very few XRN2 eCLIP reads in both HepG2 and K562 cells (Fig 3D, lower panel). We also analyzed XRN2 eCLIP binding to tRNA and miRNA genes at a global level. XRN2 binding was highest on the 3' halves of tRNAs, near the beginning of tRNA trailers for both K562 cells and HepG2 cells (S3B and S3D Fig). There was no clear pattern of binding of XRN2 to miRNAs (S3C and S3E Fig). Further evidence for XRN2 binding to tRNA trailers was recently published in Cortazar et al 2022, while our manuscript was under review. Cortazar et al 2022 show a strong peak for catalytic dead XRN2 at tRNA trailers [47]. Together, these data support a direct role for XRN2 binding to precursor tRNAs and subsequent degradation of tRF-1s.

Finally, we tested the role of the XRN2 catalytic activity in regulating tRF-1 levels. To do this, we determined tRF-1 levels in 293 cells with constitutive shXRN2 expression and either doxycycline inducible wildtype or catalytic dead (D235A) XRN2 [28]. Cells with constitutive knockdown of XRN2 expressed higher levels of tRF-1001 compared to 293T cells with intact XRN2 expression (Fig 3E). tRF-1001 was reduced below baseline upon overexpression of wild-type XRN2, while tRF-1001 increased when a catalytic dead form of XRN2 was expressed. Expression of shXRN2 and D235A XRN2 resulted in an even higher accumulation of tRF-1001 due to the additive effects of this condition. D235A XRN2 is a dominant negative protein [28], therefore overexpressing D235A XRN2 will further decrease the activity of any endogenous XRN2 that persists after shXRN2. Thus, XRN2 catalytic activity is required for XRN2 mediated degradation of tRF-1s.

## XRN2 specifically blocks tRF-1s from entering RISC

Some classes of tRFs, particularly tRF-3s, have been identified in argonaute protein complexes and have been shown to reduce gene-expression via a miRNA-like mechanism [6,26]. tRF-1s, however, are generally absent or found at low levels in argonaute proteins [24,26]. We hypothesized that XRN2 mediated degradation prevents tRF-1s and potentially other small, single stranded RNAs such as intron-tRFs, from entering Ago2. To test this hypothesis, we immunoprecipitated Flag-HA tagged Ago2 (Flag-Ago2) after knocking down XRN2 in 293T cells and performed small RNA sequencing. First, we checked the amount of overlap in significant small RNA changes between siXRN2 input and siXRN1/2 from Fig 2. Of the 72 small RNAs that significantly increased in siXRN2 input, 31 overlapped with the small RNAs that significantly increased in siXRN1/2 small RNA sequencing (Fig 4A). Out of the 31 small RNAs that significantly increased in both datasets, one was a miRNA, one mapped to a piRNA locus, one was a misc-tRF, three were intron-tRFs, and twenty-five were tRF-1s. Only two small RNAs decreased in siXRN2 input and siXRN1/2. These data suggest that XRN2 alone was primarily responsible for tRF-1 instability.

After confirming that depletion of XRN2 alone increases tRF-1s and not miRNAs in our input data (S2A and S2B Fig), we determined whether XRN2 depletion significantly alters small RNAs that bind Ago2 (S3 Table). Flag-Ago2 RNA immunoprecipitates pulled down miRNAs as expected (S2C Fig), indicating that our Ago2 immunoprecipitation was successful. tRF-1s were globally increased in the Flag-Ago2 bound fraction (Fig 4B). We note that tRF-1001 (tRF-1-tRNA-Ser-TGA-1-1) and tRF-1015 (tRF-1-tRNA-Cys-GCA-2-3) are among the tRFs that had the highest mean expression and show a large fold change in the Ago2 IP following siXRN2 (Fig 4C).

When separately examining classes of short RNAs, tRF-1s, tRF-leaders, and intron-tRFs significantly increase in Ago2 association following XRN2 depletion, while tRF-5s were

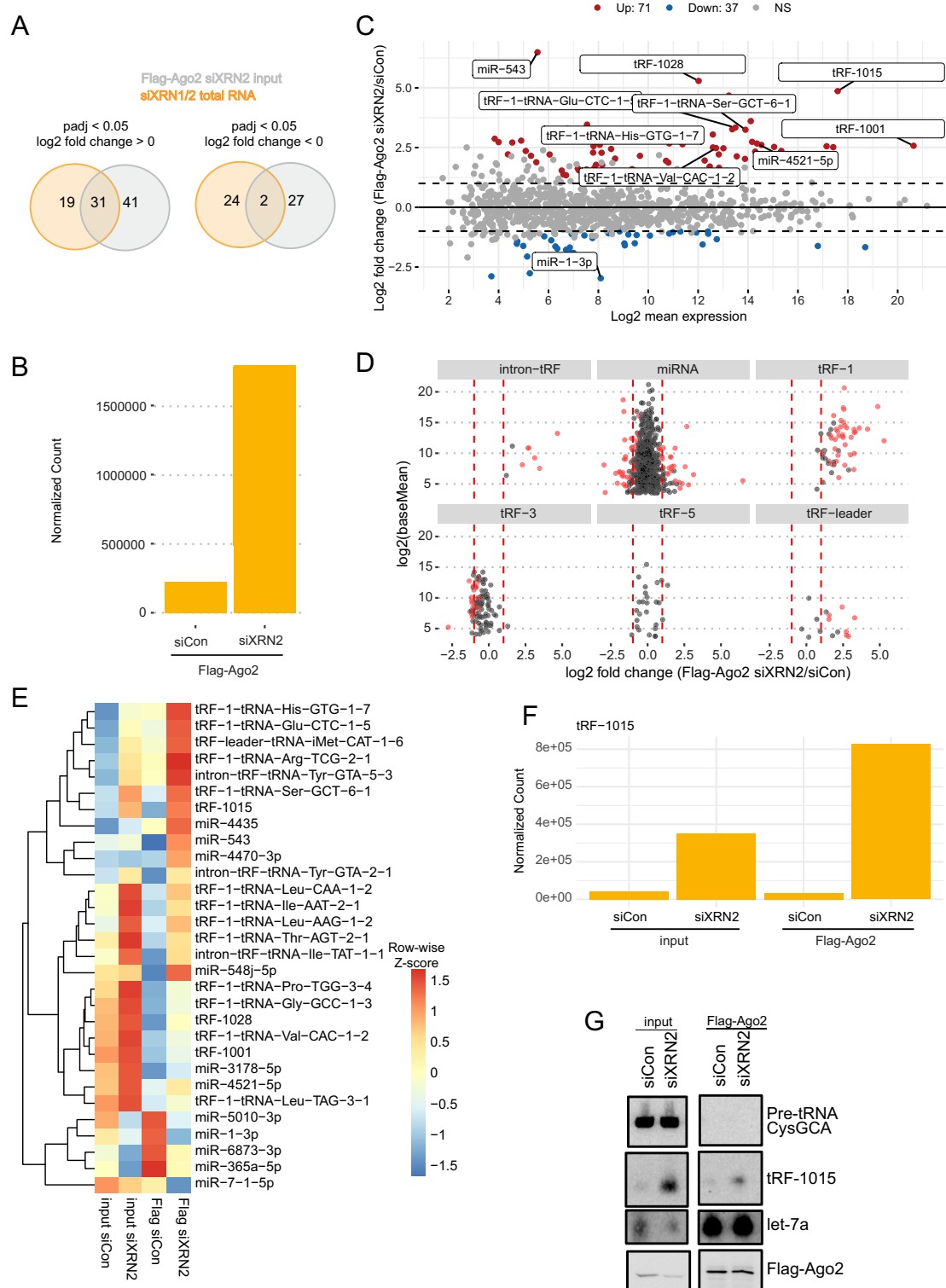

**Fig 4. XRN2 depletion leads to an accumulation of tRF-1s, in-tRFs, and tRF-leaders in Ago2.** A) Overlap of significant alterations in siXRN1/2 total RNA and Flag-Ago2 input siXRN2 from small RNA sequencing. B) Barplot of total tRF-1s in Flag-Ago2 immunoprecipitation. C) MA plot of small RNA alterations in Flag-Ago2 siCon compared to Flag-Ago2 siXRN2. Red points are significantly increased small RNAs, blue points are significantly decreased small RNAs. Differential expression determined by DESeq2. D) Significant alterations in Flag-Ago2 siXRN2 compared to Flag-Ago2 siCon by small RNA class. Red points are

significantly increased small RNAs. E) Heatmap of top 30 differentially expressed small RNAs in Flag-Ago2 siXRN2 relative to Flag-Ago2 siCon. Top 30 differentially expressed small RNAs are based on the DESeq2 adjusted p-value. A row-wise z-score transformation was performed. F) Barplot of tRF-1015 from Flag-Ago2 immunoprecipitation. G) Northern blot validation of tRF-1015 Ago2 association.

generally unaffected (Fig 4D). A few tRF-3s significantly decreased in Ago2 association upon XRN2 depletion, but the fold changes were smaller than the fold change increase seen for tRF-1s, intron-tRFs, and tRF-leaders. A small number of miRNAs are altered in Ago2 following siXRN2, but there was no consistent trend of increase or decrease of the association (Fig 4D). Next we determined the enrichment of the top 30 significantly altered sRNAs in Ago2 RIP versus input. Several tRF-1s and intron-tRFs were enriched in Ago2 relative to input after XRN2 depletion (Fig 4E). For example, tRF-1015 went from 0.5 fold enrichment in control conditions to 1.6 fold enrichment upon loss of XRN2. We also used a Northern blot to validate the increased association of tRF-1015 in Ago2 after siXRN2 (Fig 4G). Together, these data support the hypothesis that XRN2 is necessary to prevent the aberrant accumulation of tRF-1s and potentially other single stranded small RNAs in RISC.

### XRNs decrease tRF-1s and prevent tRF-1 entry into ago in Arabidopsis thaliana

XRN2 is a highly conserved and essential enzyme. The plant *Arabidopsis thaliana* (*A. thaliana*) has two nuclear XRNs, XRN2 and XRN3 (XRN2/3). Interestingly, in yeast and plants, XRNs can be inhibited by the toxic metabolite 3′-phosphoadenosine 5′-phosphate (PAP), which accumulates as a by-product of the sulfur assimilation pathway [48–50]. PAP is converted to AMP and inorganic phosphate by FIERY1 (fry1) in *A. thaliana*. We hypothesized that fry1 mutant (fry1mut) plants will accumulate tRF-1s in Ago1/2 due to inhibition of the XRN2/3 by PAP (Fig 5A). Using small RNA sequencing data from You et al. 2019, we focused on the relationship between tRF-1s, -5s, -3s, -leaders, and misc-tRFs and XRN inhibition in *A. thaliana*. Very few intron-tRFs were detected in the wildtype small RNA libraries. Manual inspection of the few introns that were greater than or equal to 14nts identified a total of 3 reads for the two wildtype small RNA sequencing libraries, perhaps due to 5′-hydroxyls and 3′ cyclic phosphates preventing cloning of more intron-tRFs during library preparation. Regardless, we found that tRF-1s are the most globally increased tRF class in two different fry1mut insertion lines (Fig 5B and 5C). tRF-1s were also increased in Ago1 and Ago2 IPs in the fry1mut lines. Only tRF-1s were globally increased in Ago1 RIP when comparing fry1mut 1–6 to wildtype (Fig 5D), with 36 tRF-1s showing significant accumulation in Ago1 RIP (Fig 5E). tRF-1, tRF-leaders and tRF-3s were all increased in the Ago2 RIP when comparing fry1mut to wt (Fig 5F), with 22 tRF-1s showing significant accumulation in Ago2 RIP (Fig 5G). These results are consistent with a role of XRNs in degrading tRF-1s and excluding them from Ago in plants, suggesting that this regulatory mechanism is important for controlling the specificity of RISC in at least two different species.

### Discussion

miRNAs compete with great precision and efficacy in a cellular environment crowded with other small RNAs of similar size and abundance. Although the miRNA effector pathway permits entry of non-miRNA small RNAs, like tRF-3s, it is unclear how other abundant small RNAs are excluded in mammalian cells. The mechanism of suppression of aberrant sRNA entry into RISC has been studied in the model organism *A. thaliana*, and involves 3′-nucleotidase FIERY1 (Fry1) and 5′ to 3′ exoribonucleases (XRNs) [50–54]. Plants and worms are

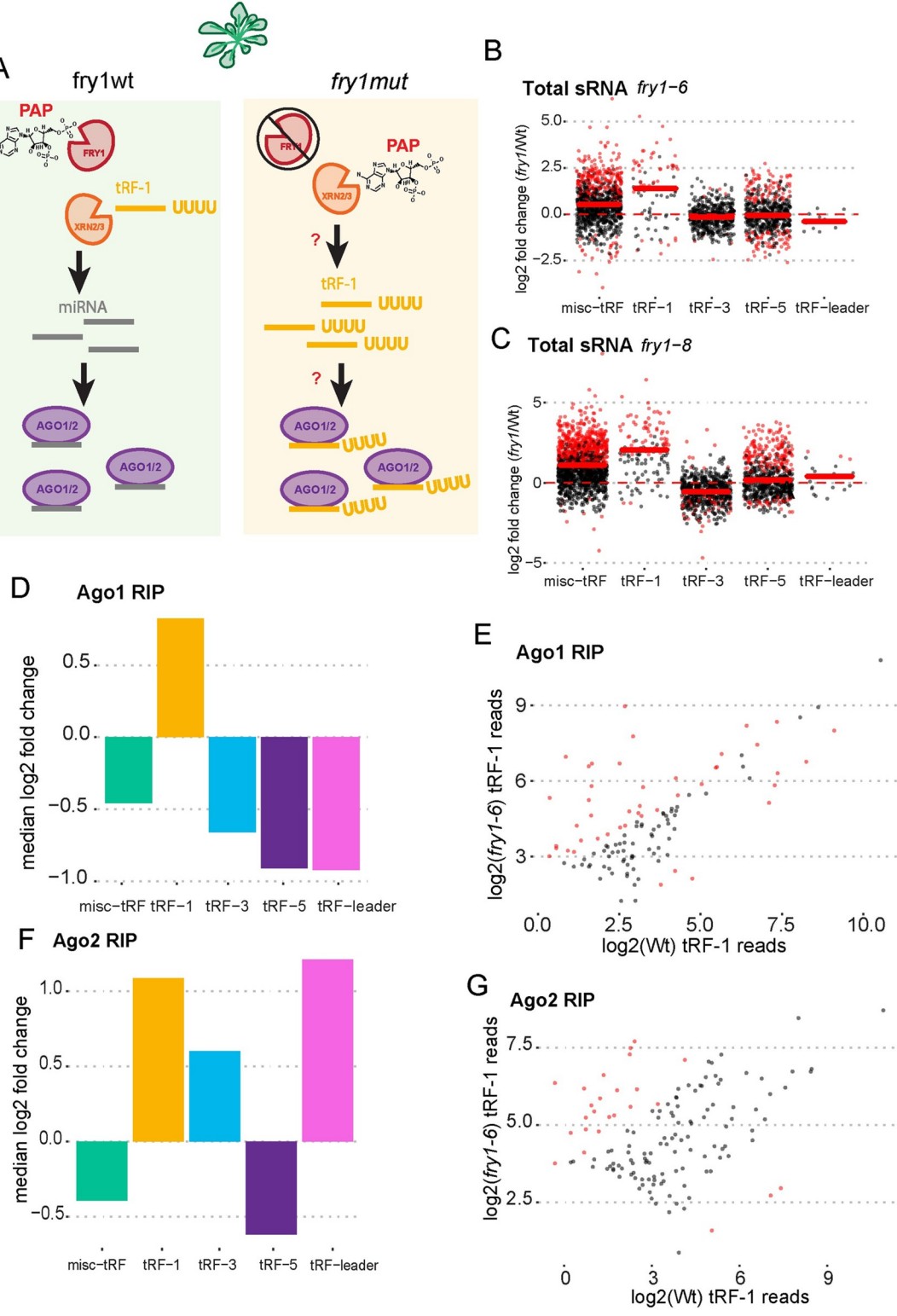

**Fig 5. *Arabidopsis thaliana* FRY1 prevents inhibition of XRN2/3, which leads to reduced tRF-1 expression and prevents tRF-1 Ago1/2 entry.** A) Schematic of hypothesis: Normally, fry1wt plants degrade PAP, ensuring the proper function of XRN2/3. tRF-1 levels are kept low in this condition in both total and Ago1/2. In fry1 mutant plants, PAP accumulates and inhibits XRN2/3. tRF-1 levels increase in total RNA fractions and Ago1/2 immunoprecipitates. Chemical structure drawn using PubChem Sketcher [88] B,C) Small RNA alterations by class in fry1mut vs fry1wt plants. Red points are statistically

significant. D,F) Median log2 fold change of fry1mut vs fry1wt Ago1 and Ago2 RNA immunoprecipitation followed by RNA sequencing. E,G) Scatter plot of tRF-1 expression in fry1wt and fry1mut plants after Ago1 and Ago2 RIP. Red points are statistically significant.

capable of producing endogenous small interfering RNAs (siRNAs) via mechanisms distinct, and absent, from mammalian miRNA biogenesis [55]. Loss of the Fry1-XRN regulatory axis leads to aberrant production of siRNAs from endogenous RNAs. However, since mammalian cells lack siRNA biogenesis machinery, it is unclear how mammalian cells prevent aberrant entry of non-miRNA small RNAs, such as tRF-1s, into RISC. It is also unknown whether plants and worms utilize the Fry1-XRN axis to prevent tRF-1 entry into the miRNA effector pathway.

Little insight into the exclusion of tRF-1s or other abundant non-miRNA small RNAs has been provided to date, but studies suggest the activity of cellular surveillance mechanisms to maintain the selectivity of RISC, perhaps similar to that in plants. Recently, one study highlights the importance of SSB/La in preventing the accumulation of aberrant tRFs within RISC [56]. SSB/La is an RNA binding protein that binds pre-tRNA trailer regions to promote their removal [57–60]. Loss of SSB/La pre-tRNA binding led to specific tRNAs entering the miRNA-effector pathway, presumably due to the formation of an alternative hairpin structure. However, this study did not address the mechanism by which tRF-1s, produced by RNaseZ in a Drosha and Dicer independent manner, avoid the miRNA-effector pathway.

In this study, we found that tRF-1s are likely excluded from RISC due to a very short half-life mediated by XRN2 activity (Fig 6). Very few tRF-1s are present at the 12 hour actD time point in HDMYZ cells, and there aren't any clear patterns in regards to tRF-1s with higher stability relative to less stable tRF-1s. Perhaps these tRF-1s are bound by unidentified proteins, interact with other RNAs, or are otherwise somehow protected from fast XRN2 turnover. The specificity of XRN2 for tRF-1s likely arises from the fact that tRF-1s are single stranded and have a 5' phosphate. Additionally, while our manuscript was under review, David Bentley and

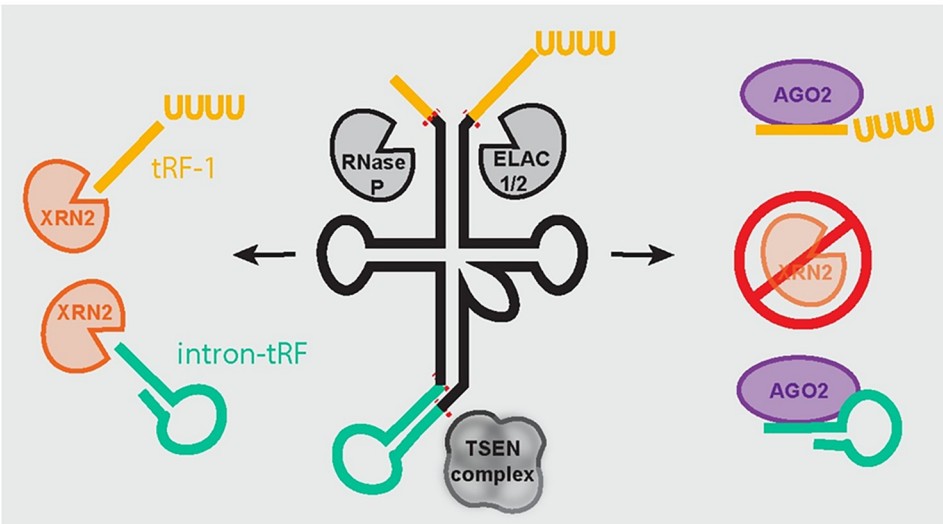

**Fig 6. Proposed model for XRN2 mediated prevention of tRF-1 entry into the miRNA effector pathway.** In mammalian cells, tRF-1s are rapidly degraded by XRN2 which prevents tRF-1 entry into the miRNA effector pathway. intron-tRFs are also significantly increased when XRN2 is deleted in mammalian cells, suggesting that XRN2 may also prevent their accumulation in Ago2. We find evidence of a similar mechanism in *A. thaliana*, where XRN inhibition by PAP accumulation results in an increase in tRF-1s associated with Ago1 and Ago2.

colleagues showed that XRN2 associates with tRNA chromatin and degrades tRNA trailers, supporting our findings [47]. Despite previous literature that suggests a role for XRN2 in miRNA turnover in C. elegans [40], the conditions in our experiments did not reveal a large global increase in miRNA levels after siXRN1/2. These data suggest either a cell- or species-specific role for XRN2 mediated regulation of miRNAs. However, implicating a broad role for XRN2 in regulating tRF-1s, we have shown that XRN2 regulates tRF-1s across many cell types and in both humans and plants.

Due to the mechanism by which tRF-1s are produced, tRF-1s are likely to be single stranded RNAs (ssRNAs). tRF-1 instability and exclusion from RISC is therefore akin to the instability of synthetic single stranded siRNAs which are known to be unstable absent the addition of modifications [61–63]. RNA modifications may be another unexplored avenue of tRF stability regulation. Interestingly, tRNAs are the most highly modified sRNAs in the cell [64], and tRFs can inherit parental tRNA modifications [27,65,66]. Although tRF-1s are not known to be modified, the presence of modifications on tRF-5s and tRF-3s may contribute to their higher stability. The higher stability of tRF-5s and tRF-3s may also explain their association with Ago proteins and the ability of tRF-3s to repress targets.

It has been hypothesized that RISC specificity is due to Dicer processing or hairpin formation [67], however, we found that tRFs can readily enter RISC if they are stabilized, regardless of Dicer processing or hairpin formation. We established that tRF-1 instability is a major factor contributing to their lack of Ago2 entry because stabilization by XRN2 depletion promoted tRF-1 entry into Ago2. We note that some tRF-1s are more efficiently associated with Ago2 following siXRN2. This may be observed for several reasons. One reason may be due to Ago2 sequence specificity [68]. Alternatively, some stabilized tRF-1s may bind and alter the function of pre-tRNA maturation proteins [69]. For example, tRF-1001 is bound by SSB/La, which leads to SSB/La sequestration from Hepatitis C virus (HCV) and blunts HCV protein synthesis [69]. SSB/La binding to tRF-1001 may be another mechanism that specifically precludes tRF-1001 entry into Ago2. Another reason some tRFs may be loaded into Ago2 more efficiently than others relates to tRF length. tRF-1001 is about 17–19 nucleotides in length, which may be a less than optimal length for Ago2 loading, while tRF-1015 is 23 nucleotides and loads much more efficiently. Finally, since we also see evidence of XRN2-tRF-1 association in eCLIP data, it is also possible that XRN2 may inhibit tRF-1- Ago2 association by competitive binding to tRF-1. The XRN2-tRF-1 regulatory axis defined in this work is likely evolutionarily important because we found evidence of this regulatory axis in different human cell lines and in the non-transformed plant *A. thaliana*.

Our data also supports a mechanism by which intron-tRFs are excluded from the miRNA-effector pathway in an XRN2-dependent manner (Fig 6). To date, little work in human cells has been done to elucidate any cellular role of intron-tRFs. This is likely due to two limitations of major small RNA sequencing protocols and analysis approaches: 1) linear tRNA introns have 5' hydroxyls and 2'-3' cyclic phosphates on the 3' end following excision by the TSEN complex, limiting tRNA intron cloning to those that are 5' phosphorylated during library preparation [23], and 2) tRNA introns can be circularized following excision (termed tricRNAs) and therefore not captured by common small RNA library preparation approaches or during computational analysis [70–72].

In a similar vein, almost nothing is known about whether tRNA-leaders can persist as short tRFs and enter Argonaute complexes. This is because size selection during the preparation of short RNA libraries selectively excludes tRNA-leaders, which are typically less than 15 nucleotides [58]. Since 14 nucleotide long isoforms of certain miRNAs, termed cityRNAs, are known to associate with and induce the catalytic activity of Ago3 [73], the <15 nucleotide tRF-leaders may have similar activities if they enter Ago complexes, and an enzyme like XRN2 may be

important to prevent such entry. Further research will be needed to determine if shorter tRFs derived from tRNA leaders are an abundant, endogenous pool of cityRNAs.

Our results raise several questions regarding the nature of non-miRNA RNAi, and future studies will focus on the precise effect of stabilized tRF-1s associated with Ago2. One possibility is that tRF-1s may regulate gene expression by base-pairing with mRNAs in a canonical miRNA-like mechanism. Other emerging evidence suggests tRFs can engage in nascent RNA silencing (NRS) by binding targets within introns in the nucleus [10] and/or coding regions [74]. Another possibility is that tRFs associated with Ago2 outcompete miRNAs and reduce their association with Ago2, similar to what has been found for rRNA derived siRNAs in Arabidopsis [51]. Another important question is under what conditions is tRF-1 entry into RISC permitted? One such case may be under transient cell states, such as stress. For example, it is known that XRN2 subnuclear localization is altered under heat stress [75,76]. Whether tRF-1s enter RISC and alter expression of heat stress related genes or aid in recovery from heat stress is unknown. Other conditions in which tRF-1s may enter RISC include conditions in which XRN inhibitor PAP is high, similar to what we have shown in Fry1 mutants in plants. There are two human orthologs of Fry1: IMPAD1/BPNT2 and BPNT1. IMPAD1/BPNT2 is localized to the golgi and autosomal recessive mutations in this gene have been shown to cause chondrodysplasia in humans [77–79]. BPNT1 has not been linked to any genetic disorders to date, however, Bpnt1-/- mice develop severe liver damage which leads to whole-body edema and death [80]. Whether XRNs are inhibited in IMPAD1/BPNT2 and BPNT1 depleted conditions is unknown, but it is possible that under these conditions tRF-1s are stabilized and contribute to the observed pathologies. Some tRF-1s may also regulate genes associated with suppression of cellular motility because XRN2 has been shown to be required for cellular invasion in glioblastoma cell lines [81]. Finally, tRF-1s may enter Ago2 more efficiently when global levels of miRNAs are diminished. miRNA levels have been shown to be depleted in DICER1 syndrome, a syndrome that predisposes patients to a variety of different malignancies [82–87]. We have previously shown that tRF-3 mediated repression of a luciferase reporter is enhanced under DICER1 depletion [6]. Although the exact effect of tRF-1 entry into Ago2 is not known, overall our data shows that tRF-1s are highly unstable, preventing their entry into the miRNA effector pathway in both human cell lines and plants.

## Supporting information

**S1 Fig. A) Length distributions of all small RNAs by actD time point.** B) tRF-1s are an abundant class of sRNAs at steady state. Shown are the normalized count distributions for each small RNA class at each actD time point in HDMYZ cells. tRF-1s are most abundant at time of generation (median abundance: 388.61 RPM). C) Decay curve for miR-21-5p in HDMYZ cells (sRNA-seq).
(TIF)

**S2 Fig. A,B) Bar plot of total number of tRF-1s (A) and miRNAs (B) in input siCon and siXRN2 conditions.** C) Bar plot of total number of miRNAs associated with Ago2 after siCon or siXRN2
(TIF)

**S3 Fig. A) Northern blots for tRF-1s after siXRN1 or siXRN2 followed by actD treatment for up to 8 hours.** Northern probes are complementary to the tRF-1 sequence. B,C) ENCODE XRN2 eCLIP metagene plot for tRNA and miRNA genes, respectively in K562 cells. D,E) Same as in B & C, except in HepG2 cells.
(TIF)

**S1 Table. Small RNAs and their half-lives from HDMYZ cells.**
(XLSX)

**S2 Table. DESeq2 analysis results for small RNA sequencing after siXRN1/2 in 293T cells.**
(XLSX)

**S3 Table. Normalized counts for small RNAs in input and Flag-Ago2 immunoprecipitation from siControl and siXRN2 293T cells.**
(XLSX)

## Acknowledgments

We would like to thank Dr. David Bentley for XRN2 mutant cells and the Genome Analysis and Technology Core at the University of Virginia for sequencing support.

## Author Contributions

**Conceptualization:** Briana Wilson, Anindya Dutta.

**Data curation:** Briana Wilson, Zhangli Su.

**Formal analysis:** Briana Wilson.

**Funding acquisition:** Briana Wilson, Zhangli Su, Anindya Dutta.

**Investigation:** Briana Wilson, Zhangli Su.

**Methodology:** Briana Wilson, Zhangli Su, Pankaj Kumar.

**Project administration:** Pankaj Kumar, Anindya Dutta.

**Resources:** Anindya Dutta.

**Software:** Pankaj Kumar.

**Supervision:** Anindya Dutta.

**Validation:** Briana Wilson.

**Visualization:** Briana Wilson, Zhangli Su.

**Writing – original draft:** Briana Wilson.

**Writing – review & editing:** Zhangli Su, Pankaj Kumar, Anindya Dutta.

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
