## [Decision Letter · Decision Letter 0]

20 Jan 2023

Dear Dr Dutta,

Thank you very much for submitting your Research Article entitled 'XRN2 Suppresses Aberrant Entry of tRNA Trailers into Argonaute in Humans and Arabidopsis' to PLOS Genetics.

The manuscript was fully evaluated at the editorial level and by independent peer reviewers. The reviewers appreciated the attention to an important problem, but raised some substantial concerns about the current manuscript. Based on the reviews, we will not be able to accept this version of the manuscript, but we would be willing to review a much-revised version. We cannot, of course, promise publication at that time.

Should you decide to revise the manuscript for further consideration here, your revisions should address the specific points made by each reviewer, particularly the comments about data replication and statistical analysis. We will also require a detailed list of your responses to the review comments and a description of the changes you have made in the manuscript.

If you decide to revise the manuscript for further consideration at PLOS Genetics, please aim to resubmit within the next 60 days, unless it will take extra time to address the concerns of the reviewers, in which case we would appreciate an expected resubmission date by email to plosgenetics@plos.org.

We are sorry that we cannot be more positive about your manuscript at this stage. Please do not hesitate to contact us if you have any concerns or questions.

Yours sincerely,

Xuemei Chen

Consulting Editor - PLoS Genetics

PLOS Genetics

Wendy Bickmore

Section Editor

PLOS Genetics

Reviewer's Responses to Questions

**Comments to the Authors:**

Reviewer #1: In this manuscript, the authors analyzed the mechanism degrading tRNA trailers (tRF-1s), which are byproducts of tRNA processing and have size similar to that of miRNAs. The authors found that the tRF-1s have short half-life and depletion of the 5’ to 3’ exoribonuclease XRN2 and XRN1 increases the stability of tRF-1s in human cell lines. In addition, knockdown of XRN2 is sufficient to increase the half-life of tRF-1s. Induced expression of wild-type XRN2, but not inactive XRN2, decreases the stability of tRF-1s in the knockdown line of XRN2. Moreover, XRN2 binds some tRF-1s. These data show that XRN2 degrades tRF-1s. The authors further showed that the amount of tRF-1s are increased in AGO IPs in the knockdown lines of XRN2 relative to wild-type. Similarly in Arabidopsis, defection in XRNs also increases the abundance of tRF-1s in total RNAs, and in AGO1/2-IPs. Based on these observations, the authors proposed that XRN2 degrades tRF-1s to exclude them from AGOs. The finding that XRN2 degrades tRF-1s is novel.

Some suggestions are:

It seems that tRF-1001 is more abundant other tRF-1s (instruction) in wild-type. why this one cannot be loaded into AGOs in wild-type, although knockdown XRN2 leads to its accumulation in AGO?

Figure 1A, why counts are less at 4 and 8 hours than at 12 hours? Please provided detailed description in legends.

Fig1b, tRF-leader also has short half-life, Does XRN2 affect its half-life? The tRF-leader is missing in fig 2C.

Fig 3c, is this result statistically significant?

polIII should be Pol III? Please go through the manuscript

Fig 4E legends, should be “altered changes in siXRN2 relative to siCons?”.

Fig 6, the model proposed XRN2 affects both tRF-1s and tRF-introns. Do the authors have data support from Arabidopsis? A brief description for the model should be provided.

Reviewer #2: The authors hope to demonstrate that XRN2 selectively degrade tRF-1s thus blocks tRF-1 accumulation in RISC in human and Arabidopsis. While the manuscript provides some interesting observations, I find some the data rudimentary and not strong enough to support the conclusion. I provide some comments below for authors’ improvement

In Fig.1B, what does each dot mean? Does each dot represent a specific small RNAs? There should be much more sequences than the dots show. some tRF-1s might be resilient than others? What is the tRNA origin of them? are there any pattern? Also, is there no replication for all the sequencing data? if this is true this is unacceptable.

The overall length distribution of different types of small RNAs should be shown, also, what is the effect of actinomycin D on the overall length distribution of all small RNAs? Will there be an overall decreased length of small RNAs? And if a small sequence become shorter, to what extent it will not be counted as a small RNA? These essential data are missing. Suppl Fig.1 showed only one sequencing result? No replication? The sequencing result after actinomycin D treatment should also be shown.

For Fig.2A, Fig.3A-C, the Northern blot data of tRNA and tRF using one probe should not be shown separately, need to be shown on a same exposure to see the relatively intensity. The authors want to hide something? Also, this is a key piece of data showing that XRN1/2, XRN2 depletion increases tRF-1s expression, with this level of quality, it will not support the conclusion. More tRF-1s shown by bioinformatic analysis (e.g., Fig.1b) need to be validated using Northern blot.

Are there any clues what XRN2 selectively degrade tRF-1s? any sequence specificity? Or due to internal/terminal modifications?

Reviewer #3: Please see the attachment.

**Have all data underlying the figures and results presented in the manuscript been provided?**

Reviewer #1: Yes

Reviewer #2: Yes

Reviewer #3: Yes

PLOS authors have the option to publish the peer review history of their article (what does this mean?). If published, this will include your full peer review and any attached files.

Reviewer #1: No

Reviewer #2: No

Reviewer #3: No

---

## [Decision Letter · Decision Letter 1]

21 Apr 2023

Dear Dr Dutta,

We are pleased to inform you that your manuscript entitled "XRN2 Suppresses Aberrant Entry of tRNA Trailers into Argonaute in Humans and Arabidopsis" has been editorially accepted for publication in PLOS Genetics. Congratulations!

Yours sincerely,

Wendy A. Bickmore

Section Editor

PLOS Genetics

Wendy Bickmore

Section Editor

PLOS Genetics

Comments from the reviewers (if applicable):

Reviewer's Responses to Questions

**Comments to the Authors:**

Reviewer #1: My concerns have been addressed.

Reviewer #2: My concerns/questions have been addressed

Reviewer #3: The big problem is the figure quality for the revised manuscript. Please make sure to provide high resolution figures for publication. If not, I would like to reject this manuscript.

**Have all data underlying the figures and results presented in the manuscript been provided?**

Reviewer #1: Yes

Reviewer #2: Yes

Reviewer #3: Yes

PLOS authors have the option to publish the peer review history of their article (what does this mean?). If published, this will include your full peer review and any attached files.

Reviewer #1: No

Reviewer #2: No

Reviewer #3: No

**Data Deposition**

http://datadryad.org/submit?journalID=pgenetics&manu=PGENETICS-D-22-01273R1

**Press Queries**

---

## [Editor Report · Acceptance letter]

2 May 2023

PGENETICS-D-22-01273R1 

XRN2 Suppresses Aberrant Entry of tRNA Trailers into Argonaute in Humans and Arabidopsis 

Dear Dr Dutta, 

We are pleased to inform you that your manuscript entitled "XRN2 Suppresses Aberrant Entry of tRNA Trailers into Argonaute in Humans and Arabidopsis" has been formally accepted for publication in PLOS Genetics! Your manuscript is now with our production department and you will be notified of the publication date in due course.

With kind regards,

Zsofi Zombor

PLOS Genetics

On behalf of:
